# The Genus *Allochrusa*: A Comprehensive Review of Botany, Traditional Uses, Phytochemistry, and Biological Activities

Rano Mamadalieva [1,*], Vahobjon Khujaev [1], Michal Šoral [2], Nilufar Z. Mamadalieva [3,4] and Michael Wink [5]

[1] Kokand State Pedagogical Institute, Turon Str. 23, Kokand 713000, Uzbekistan; xujayev_030@mail.ru

[2] Analytical Department, Institute of Chemistry, Slovak Academy of Sciences, Dúbravská Cesta 9, SK-845 38 Bratislava, Slovakia; michal.soral@savba.sk

[3] Tashkent Institute of Irrigation and Agricultural Mechanization Engineers, National Research University, Kori Niyazov Str. 39, Tashkent 100000, Uzbekistan; nmamadalieva@yahoo.com

[4] Institute of the Chemistry of Plant Substances, Uzbekistan Academy of Sciences, M. Ulugbek Str. 77, Tashkent 100170, Uzbekistan

[5] Institute of Pharmacy and Molecular Biotechnology, Heidelberg University, 69120 Heidelberg, Germany; wink@uni-heidelberg.de

\* Correspondence: rmamadalieva@yahoo.com

**Abstract:** The genus *Allochrusa* (Caryophyllaceae) comprises nine species, which are native to Central Asia, Turkey, Iran, Afghanistan, and the Caucasus. They have been used in folk medicine and in the preparation of various sweets and detergents, especially in Asian countries. A diversity of secondary metabolites has been reported from the genus *Allochrusa*, including triterpene glycosides, ecdysteroids, flavonoids, volatile compounds, fatty acids, polysaccharides, pectins, hemicelluloses, and other phytochemicals. In vitro and in vivo pharmacological studies on isolated compound fractions and extracts from *Allochrusa* species showed anti-inflammatory, adjuvant, hemolytic, cytotoxic, antifungal, analgesic, antioxidant, and other activities. In this review, the chemical compounds and diverse biological activities of the *Allochrusa* genus are summarized.

**Keywords:** *Allochrusa gypsophiloides*; saponin; triterpene glycosides; foods; biological activities

## 1. Introduction

The Caryophyllaceae, also referred to as the "pink or carnation family", includes approximately 80 genera with more than 2600 species [1]. The family is famous for widely used genera including *Silene*, *Dianthus*, *Gypsophila*, *Stellaria*, *Saponaria*, and *Allochrusa*. Members of the Caryophyllaceae include important medicinal, ornamental, and aromatic plants and have been used as traditional herbal medicines and in various branches of industry [2]. *Agrostemma githago* (Corncockle), *Dianthus chinensis* (Pink), *D. caryophyllus* (Clove pink), *D. barbatus* (Sweet William), *Gypsophila paniculata* (Baby's Breath), *Lychnis coronaria* (Rose campion), *Saponaria officinalis* (Soapwort), *S. ocymoides* (Rock soapwort), and *Silene* spp. (Campions) are known as ornamental species. These plants are famous in the cut flower trade for their beautiful flowers. Caryophyllaceae are known to be a rich source of pharmacologically active secondary metabolites. The major chemical constituents of this family are saponins, flavonoids, ecdysteroids, sterols, lignans, polyphenols, essential oils, and N-containing compounds such as vitamins, alkaloids, and cyclic peptides [3]. Some saponin-containing species from the Caryophyllaceae, such as *Saponaria officinalis* (soapwort), *Gypsophila paniculata* (Baby's breath), and *Allochrusa gypsophiloides* (Turkestan soap root), have been used as soap since time immemorial. *Silene*, *Acanthophyllum*, *Gypsophila*, *Dianthus*, *Stellaria*, and *Saponaria* are the most studied genera for both ethanomedicinal and biological studies. Their utilizations in traditional medicine include anticancer, anti-inflammatory, antifungal, antiviral, antibacterial, and antioxidant properties [4].

The pink family contains three subfamilies called Paronychioideae, Alsinoideae, and Caryophylloideae, according to the presence or absence of stipules as well as the type of corolla and calyx [1,3]. Formerly, species of the genus *Allochrusa* were included in the genus *Acanthophyllum*; they comprise about nine species distributed in Turkey, Central Asia, the Caucasus, and Iran. The Plant List recognizes nine accepted species: *Allochrusa bungei* Boiss., *A. paniculata* (Regel & Herder) Ovcz. & Czukav, *A. gypsophiloides* (Regel) Schischk. [5], *A. persica* (Boiss.) Boiss., *A. tadshikistanica* Schischk., *A. takhtajanii* Gabrieljan & Dittrich, *A. transhyrcana* Czerep., *A. versicolor* (Fisch. & C. A. Mey.) Boiss. (http://www.theplantlist.org/1.1/browse/A/Caryophyllaceae/Allochrusa/, accessed on 20 November 2022).

The aims of this study are to compile all the most recent information available on *Allochrusa* plants and provide detailed yet concise references dealing with distribution, taxonomy and botanical characterization, traditional uses, primary and secondary metabolites, and pharmacological activities. This review sheds light on this less known plant section that might attract researchers in drug discovery to identify novel compounds with potential biological properties.

## 2. Materials and Methods

The literature for this review was collected by searching various scientific electronic databases, including PubMed (https://pubmed.ncbi.nlm.nih.gov/, accessed on 12 December 2022), GoogleScholar (https://scholar.google.com/, accessed on 12 December 2022), SciFinder (https://scifinder.cas.org/, accessed on 12 December 2022), SpringerLink (https://link.springer.com/, accessed on 12 December 2022), ScienceDirect (https://www.sciencedirect.com/, accessed on 12 December 2022), and Web of Science (https://mjl.clarivate.com/, accessed on 12 December 2022), via searching the following keywords: "*Allochrusa* and phytochemistry", "*Allochrusa* and chemical composition", "*Allochrusa* and traditional uses", and "*Allochrusa* and biological activity" in the literature written in English, Russian, and Uzbek. We also searched the references for the former name *Acanthophyllum sp.* (e.g., for *Acanthophyllum gypsophiloides*, *Acanthophyllum panuculatum*, etc.). Additional information from other sources was extracted (books, theses). All references covering the isolation of the metabolites, as well as the taxonomy, botanical characterization, occurrence, traditional uses, and pharmacological properties of the *Allochrusa* species from 1965 to 2022, were evaluated for this review. The Plant List (http://www.theplantlist.org/, accessed on 23 December 2022), Plants of the World Online (https://powo.science.kew.org/, accessed on 23 December 2022), and International Plant Name Index (https://ipni.org/, accessed on 23 December 2022) databases were used to validate the scientific names.

In this review, we start by describing the general botanical characteristics of species in the genus *Allochrusa*, followed by an overview of the distribution of the species worldwide and a compilation of traditional uses. Then, the isolated secondary metabolites are grouped in subclasses based on their chemical structures, and finally the traditional uses of these plants are discussed.

## 3. Taxonomy and Diversity

The genus *Acanthophyllum* Meyer (Caryophyllaceae) in a broad sense comprises 80–90 perennial semi-shrub species that are primarily found in the Irano-Turanian region [6]. *Acanthophyllum* consists of five sections: *Macrodonta* (Boissier) Pax in Engler & Prantl, *Macrostegia* (Boissier) Pax in Engler & Prantl, *Pleiosperma* (Boissier) Pax in Engler & Prantl, *Oligosperma* Schischkin ex Schiman-Czeika, and *Allochrusa* Bunge ex Boiss. Earlier investigations of molecular phylogeny supported that species of *Allochrusa* share a clade together with *Acanthophyllum*. Because of significant morphological differences, *Allochrusa* has been treated recently as a separate genus [7] (http://www.theplantlist.org/1.1/browse/A/Caryophyllaceae/Allochrusa/, accessed on 12 December 2022).

*Allochrusa gypsophiloides* (syn. *Acanthophyllum gypsophiloides*) is a perennial plant. The stem is strongly branched and has short hairs or is smooth. Due to the numerous branching of its stems, the plant has a spherical shape and its height and diameter reach almost

the same 70–80 cm. In contrast, the leaves are up to 0.5 cm wide and 3 cm long, flat, almost linear or linear-lanceolate, with a pointed tip. The sepals are lanceolate, smooth, and 2-3 times shorter than the petals. The flowers are located in free, forked-branched, wide-grooved inflorescences at the ends of stems and branches. The middle flowers of each branch of the inflorescence are almost sessile; the other two have long filamentous inflorescences. Sepals are five-lobed, cylindrical, smooth, about 2 mm long; they consist of five pinkish-yellow leaves, the node is single-celled, and there are 10 stamens. The fruit is an indehiscent, membranous, one-to-two-seeded calyx with preserved sepals. The seeds are light brown, almost spherical, slightly flattened on the sides, ripen in early August, but do not fall until autumn. The root is an arrowroot, sometimes globose, twisted, up to 2 m long and more, weighing up to 12 kg. Lateral roots branch several times from the tap root. Roots do not have the ability to regenerate after being mined, so the natural dense growth areas of this plant are not regenerated after the raw materials are collected [8]. *A. tadshikistanica* is a perennial herbaceous plant that grows 30–60 cm high. Stems a few and thin. Leaves 10–25 mm long, linear, flat, without thorns. Flowers pale pink, in terminal lax, spreading panicle. Fruit is a capsule. Flowering in May–August, fruiting in August–September [5,9].

Species of the genus *Allochrusa* are distributed in Afghanistan, Armenia, Azerbaijan, Iran, Iraq, Kazakhstan, Kyrgyzstan, Tajikistan, Turkmenistan, Turkey, and Uzbekistan. These plants normally grow in arid regions, deserts, mountain slopes, and temperate zones [10]. In the territory of Iran, four species of the section *Allochrusa* have been found: *Allochrusa bungei*, *A. persica*, *A. versicolor*, and *A. lutea*. *A. lutea* is a recently recorded species in the flora of Iran; its natural habitat is limited to northwest Iran [11] (Table 1). *A. takhtajanii* was listed as a critically endangered species endemic to a region in Armenia in 2014 [12]; *A. transhyrcana* and *A. tadshikistanica* are endemic to Turkmenistan and Tajikistan, respectively [10]. In the flora of Kazakhstan, *A. gypsophiloides* is found in four districts of the Turkistan (formerly South Kazakhstan) region [13]. In Kyrgyzstan: The Western Tien Shan and Pamir rivers' dry slopes and abandoned arable land at an altitude of 400 to 700 m above sea level are the primary habitats for *A. gypsophiloides* and *A. paniculata* species [14]. Two species of *Allochrusa*, *A. versicolor* and *A. bungei*, have been found in Turkey [15] (Figure 1).

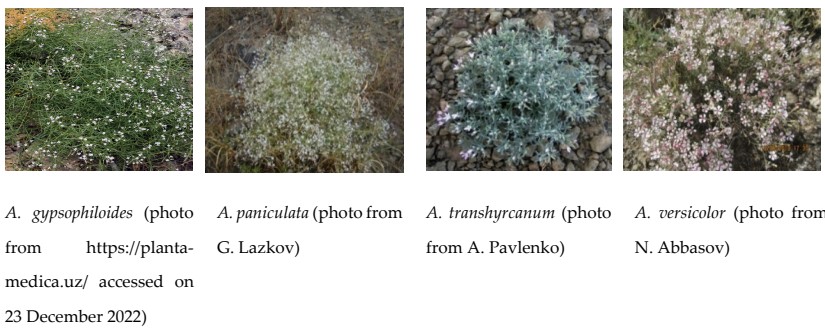

*A. gypsophiloides* (photo from https://planta-medica.uz/ accessed on 23 December 2022)

*A. paniculata* (photo from G. Lazkov)

*A. transhyrcanum* (photo from A. Pavlenko)

*A. versicolor* (photo from N. Abbasov)

**Figure 1.** Photos of some *Allochrusa* species (https://www.plantarium.ru/lang/en/page/view/item/43196.html/, accessed on 23 December 2022).

In Uzbek flora, *Allochrusa* is described by three species: *Allochrusa gypsophiloides* (syn. *Acanthophyllum gypsophiloides*, local name Etmak), *A. paniculata* (syn. *Acanthophyllum paniculatum*), and *A. tadshikistanica* (syn. *Acanthophyllum tadshikistanicum*) [16]. They are perennial herbaceous saponin-bearing plants. *A. gypsophiloides* occurs in a relatively limited area; it grows only in the western Tien Shan and Pamir-Altay on foothill desert loess steppes, dry slopes of rivers, and on gravelly slopes at 400 to 1500 m above sea level. Eisenman et al. [17] declare that the presence of *A. gypsophiloides* is more typical at altitudes of 1200–1500 m in Jizzax, Qashqadaryo, Samarqand, Surxondaryo, and Toshkent regions of Uzbekistan on stony slopes with rocky debris of the adyr and tau zones (Figure 2). Small white flowers form openwork clusters on a branched, almost spherical stem. The species is distinguished by a long lifespan (up to 200 years) and a powerful tap root, reaching a weight

of 2–12 kg. As a consequence of extensive and unsystematic harvesting, *A. gypsophiloides* has been added to the list of rare species with a strongly decreasing population in the Red Book of Uzbekistan [18]. *A. paniculata* (Regel & Herder) Ovcz. & Czukav. is a plant native to Central Asia (Dzungarian Alatau, Tian-Shan, and Alay mountains), where it grows on rocky and graveled slopes of foothills in the lower range of mountains [5,9,19,20]. *A. tadshikistanica* is a herbaceous perennial endemic to the southern Pamir-Alay and recorded in the Red Book of Uzbekistan [5,9,18].

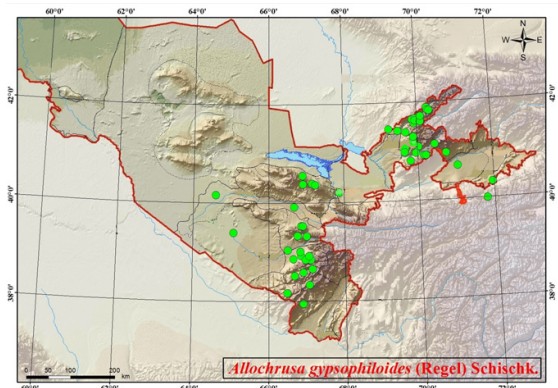

**Figure 2.** Distribution of *A. gypsophiloides* (with green dots) in the territory of Uzbekistan (https://planta-medica.uz/, accessed on 3 January 2023).

**Table 1.** Diversity and distribution of different *Allochrusa* species in Asia.

| Species | Distribution | Region | References |
|---|---|---|---|
| *A. takhtajanii, A. bungei, A. versicolor* | Armenia | Surenavan, Urts Mountains, Transcaucasus | [10]; gbif.org |
| *A. versicolor* | Azerbaijan | South to north-west Transcaucasus | [10] |
| *A. bungei, A. lutea, A. persica, A. tadshikistanica, A. versicolor* | Iran | Tabriz, Marand and Jolfa, north-west of Iran | [11] |
| *A. gypsophiloides, A. paniculata* | Kazakhstan | Karatau, Mount Kazigurt, Aksu-Zhabagly, South Kazakhstan | [17] |
| *A. gypsophiloides, A. paniculata* | Kyrgyzstan | Chatkal, Talas, and Pskem ranges; Talas Ala-Too, Kyrgiz Ala-Too | [17,18] |
| *A. gypsophiloides, A. transhyrcana* | Turkmenistan | Amu Darya river | [10,18] |
| *A. gypsophiloides, A. tadshikistanica, A. paniculata* | Tajikistan | Pamir mountains | [10,21] |
| *A. gypsophiloides* | Uzbekistan | Chatkal, Hissar, Kurama, Kuhitang ranges (Dzhizak, Kashkadarya, Namangan, Samarkand, Surkhandarya, Tashkent regions) | [17,20] |
| *A. paniculata* | | Andijan, Fergana regions | [20] |
| *A. tadshikistanica* | | Chatkal, Hissar, Kurama ranges (Kashkadarya, Namangan, Surkhandarya, Tashkent regions) | [22] |

## 4. Traditional Uses

The traditional uses of *Allochrusa* plants date back centuries. The usage of two species of *Allochrusa* is implemented in official medicine: *A. gypsophiloides* and *A. paniculata*. *A. gypsophiloides* are widely used in conventional medicine and various branches of industry. The roots of *Allochrusa* (soap root) are used in the food, pharmaceutical, construction, fur, textile, dyeing, and perfume industries. *A. gypsophiloides* is commonly known as Turkestan soaproot, Beh or Yetmak. *A. gypsophiloides* is a valuable medicinal and industrial plant that has been widely exported for a long time as the primary source of saponins. *A. gypsophiloides* (as is the case with most Caryophyllaceae) produces saponins that are utilized

in manufacturing sweets or natural washing aids [17]. *A. gypsophiloides* is characterized by high content levels of triterpene saponins (up to 30%) with high surface and hemolytic activities [23]. A Central Asian sweet delicacy with a tonic effect called "nisholda" is prepared using large quantities of the roots of *A. gypsophiloides* as a key ingredient [24]. People also use the roots for the preparation of such oriental sweetmeats as parvarda and halva. Additionally, the mining industry (for flotation and the efficient extraction of minerals from waste rocks), the fur cleaning process, and the electrolysis of metals all make extensive use of the roots of saponin-bearing plants [16]. The saponins of *A. tadshikistanica* are also used for the local production of confectionery [5].

## 5. Medicinal Uses

In traditional medicine, the decoction of the roots of *A. tadshikistanica* is used as an expectorant for bronchitis and other respiratory conditions, similar to other plant drugs with saponins [25–28]. *A. tadshikistanica* (contains saponins) has an expectorant and wound-healing effect [21]. By stimulating the Nervus vagus in the stomach, saponins indirectly induce the secretion of water in the lungs; the secretion of water solubilizes the mucus (typical for an expectorant). An infusion of roots of *A. gypsophiloides* is used as a choleretic, diuretic, and laxative in traditional medicine (a teaspoon of crushed roots is poured into a glass of cold boiled water, left to stand for 8–10 h, and is drunk during the day). Tea or a drink can be brewed to treat gastrointestinal, spleen, skin, liver, kidney, and venereal diseases as well as metabolic dysfunction. Saponins can increase the immunogenicity of several antigens and have been employed in veterinary medicine as an adjuvant for vaccines against anthrax and brucellosis [24]. *Allochrusa gypsophiloides* extracts have been found to have high immunostimulatory [29], antiviral [30], and anti-tumor action [31–34].

## 6. Phytochemical Compounds

To our knowledge, the phytochemistry of saponin-containing plants of the *Allochrusa* is yet to be investigated comprehensively. Only three species have undergone thorough research. Studies on the phytochemical content of various *Allochrusa* species have shown the presence of several classes of primary and secondary metabolites. A literature search found that the *Allochrusa* is a rich source of triterpene glycosides including saponins [35–42], polysaccharides [16,43–46], ecdysteroids [21,47], flavonoids [21,47,48], volatile compounds [49], lignins [50], and other compounds. More than 70 metabolites have been found in the genus. Among these substances, triterpene glycosides are the principal active constituents of *Allochrusa* species and a typical characteristic of Caryophyllaceae. In the following section, the isolated compounds from the *Allochrusa* species based on their chemical structures are classified. The most appropriate citation relating to the isolation and characterization of these metabolites is mentioned and highlighted.

### 6.1. Triterpene glycosides

Triterpenoid glycosides, also known as saponins, are glycosides with a remarkable variety in their structural and biological activities [51]. Historically, plant products with high saponin content have been used as detergents, for example, soapwort (*Saponaria officinalis*; Caryophyllaceae), soapbark (*Quillaja saponaria*; Rosaceae), and shikakai powder (*Gleditsia sinensis*; Fabaceae). Though triterpenoid saponins are rarely found in monocotyledons (they contain steroidal saponins), they are abundant in many dicotyledonous families such as Apiaceae, Araliaceae, Caryophyllaceae, Cucurbitaceae, Fabaceae, and Ranunculaceae [52]. Chemically, these families are distinguished by a high production of saponins, which are dominated by four oleanane-type aglycones: gypsogenin, quillaic acid, gypsogenic acid, and 16a-hydroxy-gypsogenic acid [53].

The most characteristic and prominent compounds in *Allochrusa* are terpene glycosides, the highest amount of which were described in *A. gypsophiloides* and *A. paniculata*. In previous research on *A. gypsophiloides* growing in Uzbekistan, the saponin concentration in the aerial parts was 0.79%, and that of the roots ranged from 14.19 to 22.16% depending on

their age [48,54]. Few data exist on the amount of saponins in *A. gypsophiloides* based on plant age and growth stage. Accordingly, the content of saponins in the roots rises with plant age, from 9.5% in annuals to 23% in plants that are five years old [55]. After the introduction of *A. gypsophiloides* to Turkmenistan, phytochemical studies were conducted yielding results revealing that the maximal concentration of saponins in the plant roots (up to 20%) could only be observed after five years of culture [56]. From the methanol root, the extract of *A. gypsophiloides* gypsoside (**1**) was isolated, which is similar to a triterpene glycoside previously isolated from *Gypsophila paniculata* and *Gypsophila pacifica* [35]. Later, Belous and Ryabinin [57] found gypsogenic acid (**8**) in considerably lower yields by the saponification of a methanolic extract of *A. gypsophiloides*. Allochroside is a mixture of four glycosides of triterpene saponins from *A. gypsophiloides*. It contains acanthophylloside A (minor component, chemical formula unspecified) [36], acanthophylloside B (2, 40%), acanthophylloside C (3, 25%), and acanthophylloside D (4, 25%). They belong to the group of 3,28-*O*-bidesmoside saponins, specifically glucuronides of quillaic acid and gypsogenin. Compounds from this group have a (substituted) glucuronic acid moiety at the C-3 hydroxyl of the aglycone and belong to the highest glycosylated bidesmosides. Acanthophyllosides B-D (2–4) were identified in the roots of *A. gypsophiloides* [36,37,40–42,54]. Two triterpenoid saponins, 5 and 6, were isolated by Khatuntseva et al. [58] from the methanolic extract obtained from the roots of *A. gypsophiloides*. The saponins 5 and 6 have quillaic acid or gypsogenin moieties as an aglycon, as in the case of acanthophyllosides B-D, and both bear similar sets of oligosaccharide chains—a 3-*O*-linked α-L-Arap-(1→3)-[α-D-Galp-(1→2)]-β-D-GlcpA trisaccharide and a β-D-Xylp-(1→3)-β-D-Xylp-(1→3)-α-L-Rhap-(1→2)-[β-D-Quip-(1→4)]-β-D-Fucp pentasaccharide connected through an ester linkage to C-28 (Table 2). Another triterpene glycoside, paniculatoside C (**7**), was found in the roots of *A. paniculata* [41,59,60].

### 6.2. Ecdysteroids

Ecdysteroids are steroid hormones present in insects, aquatic animals, fungi, and higher plants. Plant ecdysteroids are important antifeedant defense compounds, as they interfere with the molting hormone ecdysone of insects. Currently, there are more than 500 known structural analogues of ecdysteroids. They are produced biochemically from C27-, C28-, or C29- sterols. 20-Hydroxyecdysone (2β,3β,14α,20R,22R,25-hexahydroxycholest-7-en-6-one) is the most commonly encountered analogue, which is found in plants and arthropods [61,62]. Ecdysteroids are primarily produced in the tribe Lychnideae of the subfamily Caryophylloideae of Caryophyllaceae, in which *Silene*, *Lychnis*, *Petrocoptis*, *Sagina*, and *Saponaria* are the main ecdysteroid-synthesizing genera of the Caryophyllaceae [63,64]. *Allochrusa* species have been found to contain some ecdysteroids. An investigation of the qualitative and quantitative composition of 20-hydroxyecdysone (ecdysterone) (**9**) and polypodine B (**10**) in the aerial parts of the species *A. tadshikistanica* was carried out by Darmogray et al. [21]. The content of these ecdysteroids in this plant was determined by HPLC. Later, another representative of the ecdysteroids, 3-epi-2-deoxyecdysone (*syn.* 3α,14α,22R,25-tetrahydroxy-5β(H)-cholest-7-en-6-one) (**11**), having analgesic and anti-inflammatory activities, was obtained from a water-ethanol extract of the aerial part of *A. gypsophiloides* [47,65]. This plant was discovered to be a prospective source of the main ecdysteroid 9 whose content in the water-ethanol extract was 0.19%.

### 6.3. Flavonoids

Flavonoids are a group of polyphenolic compounds that belong to one of the most widely studied constituents of the Caryophyllaceae [3]. Although some species of *Allochrusa* were investigated chemically, the characterization of flavonoids in this genus is incomplete. Grudzinskaya et al. [48] reported that *A. gypsophiloides* and *A. paniculata* contain the flavonoid quercetin (**12**). In an HPLC analysis, the flavonoid vicenin (**13**) was first detected in *A. tadshikistanica* [21]. It should be noted that the *C*-glycosides of flavones are less common than other flavonoids.

**Table 2.** Structures and distribution of secondary metabolites in the genus *Allochrusa*.

| Plant Sources | Name | Formula | Structure | | References |
|---|---|---|---|---|---|
| | | | Triterpene glycosides | | |
| | | |  | | |
| *A. gypsophiloides* | Gypsoside (**1**) | $C_{80}H_{126}O_{44}$ | $R_1$ = -β-D-GlcUA- | 4←1-β-D-Glc<br>-4←1-β-D-Gal<br>3←1-α-L-Ara | [35] |
| | | | $R_2$ = -α-L-Rha- | 4←1-β-D-Fuc-3←1-β-D-Xyl<br>2←1-β-D-Xyl-3←1-β-D-Xyl | |
| | | | $R_3$ = -H | | |
| *A. gypsophiloides* | Acanthophylloside B (**2**) | $C_{86}H_{136}O_{48}$ | $R_1$ = -β-D-GlcUA- | 4←1-α-L-Ara<br>-4←1-β-D-Gal<br>2←1-β-D-Gal | [36–39,41,42,54] |

**Table 2.** *Cont.*

| Plant Sources | Name | Formula | Structure | | References |
|---|---|---|---|---|---|
| | | | R₂ = -β-D-Fuc- | 4←1-α-L-Rha-4←1-β-D-Xyl-3←1-β-D-Xyl-3←1-β-D-Xyl<br>2←1-α-D-Qui | |
| | | | R₃ = -H | | |
| *A. gypsophiloides* | Acanthophylloside C (**3**) | $C_{92}H_{146}O_{53}$ | R₁ = -β-D-GlcUA- | 6←1-β-D-Glc<br>4←1-α-L-Ara<br>-4←1-β-D-Gal<br>2←1-β-D-Gal | [36–39,41,42,54] |
| | | | R₂ = -β-D-Fuc- | 4←1-α-L-Rha-4←1-β-D-Xyl-3←1-β-D-Xyl<br>-3←1-β-D-Xyl<br>2←1-α-D-Qui | |
| | | | R₃ = -H | | |
| *A. gypsophiloides* | Acanthophylloside D (**4**) | $C_{86}H_{136}O_{49}$ | R₁ = -β-D-GlcUA- | 4←1-α-L-Ara<br>-4←1-β-D-Gal<br>2←1-β-D-Gal | [40,41] |
| | | | R₂ = -β-D-Fuc- | 4←1-L-Rha-<br>4←1-β-D-Xyl-<br>3←1-β-D-Xyl-<br>3←1-β-D-Xyl | |
| | | | R₃ = -OH | | |
| *A. gypsophiloides* | 3-*O*-[β-D-Galactopyranosyl-(1→2)-[α-L-arabinopyranosyl-(1→3)]-β-D-glucuronopyranosyl] gypsogenin 28-β-D-xylopyranosyl-(1→3)-β-D-xylopyranosyl-(1→3)-α-L-rhamnopyranosyl-(1→2)-[6-deoxy-β-D-glucopyranosyl-(1→4)]-β-D-fucopyranosyl ester (**5**) | $C_{75}H_{118}O_{39}$ | R₁ = -β-D-GlcUA- | 2←1-β-D-Gal<br>3←1-α-L-Ara | [58] |

**Table 2.** *Cont.*

| Plant Sources | Name | Formula | Structure | | References |
|---|---|---|---|---|---|
| | | | R$_2$ = -β-D-Fuc- | 2←1-α-L-Rha-3←1-β-D-Xyl-3←1-β-D-Xyl<br>4←1-α-D-Qui | |
| | | | R$_3$ = -H | | |
| *A. gypsophiloides* | 3-*O*-[β-D-Galactopyranosyl-(1→2)-[α-L-arabinopyranosyl-(1→3)]-β-D-glucuronopyranosyl]quillaic acid 28-β-D-xylopyranosyl-(1→3)-β-D-xylopyranosyl-(1→3)-α-L-rhamnopyranosyl-(1→2)-[6-deoxy-β-D-glucopyranosyl-(1→4)]-β-D-fucopyranosyl ester (**6**) | C$_{75}$H$_{118}$O$_{40}$ | R$_1$ = -β-D-GlcUA- | 2←1-β-D-Gal<br>3←1-α-L-Ara | [58] |
| | | | R$_2$ = -β-D-Fuc- | 2←1-α-L-Rha-3←1-β-D-Xyl-3←1-β-D-Xyl<br>4←1-α-D-Qui | |
| | | | R$_3$ = -OH | | |
| *A. paniculata* | Paniculatoside C (**7**) | C$_{54}$H$_{86}$O$_{25}$ |  | | [41,59,60] |
| | | | R = -β-D-Glc | 4←1-β-D-Glc<br>6←1-β-D-Glc-6←1-β-D-Glc | |

**Table 2.** *Cont.*

| Plant Sources | Name | Formula | Structure | References |
|---|---|---|---|---|
| *A. gypsophiloides* | Gypsogenic acid (**8**) | $C_{30}H_{46}O_5$ | R = H | [57] |
| **Ecdysteroids** | | | | |
| *A. tadshikistanica,* *A. gypsophiloides* | 20-Hydroxyecdysone (**9**) | $C_{27}H_{44}O_7$ | R = H | [21,47] |
| *A. tadshikistanica* | Polypodine B (**10**) | $C_{27}H_{44}O_8$ | R = OH | [21] |
| *A. gypsophiloides* | 3-epi-2-Deoxyecdysone (**11**) | $C_{27}H_{44}O_5$ | | [47,65] |
| **Flavonoids** | | | | |
| *A. paniculata* | Quercetin (**12**) | $C_{15}H_{10}O_7$ | | [48] |
| *A. tadshikistanica* | Vicenin (**13**) | $C_{27}H_{30}O_{15}$ | | [21] |

**Table 2.** *Cont.*

| Plant Sources | Name | Formula | Structure | References |
|---|---|---|---|---|
| | | **Major volatile compounds** | | |
| *A. gypsophiloides* | Pulegone (**14**) | $C_{10}H_{16}O$ | | [49] |
| *A. gypsophiloides* | *trans-p*-Menthan-3-one (**15**) | $C_{10}H_{18}O$ | | [49] |
| *A. gypsophiloides* | *trans*-Verbenol (**16**) | $C_{10}H_{16}O$ | | [49] |
| *A. gypsophiloides* | Phytol (**17**) | $C_{20}H_{40}O$ | | [49] |
| *A. gypsophiloides* | 2-Isopropyl-5-methylcyclohexanone (**18**) | $C_{10}H_{18}O$ | | [49] |
| *A. gypsophiloides* | Isopiperitenon (**19**) | $C_{10}H_{14}O$ | | [49] |

*6.4. Essential Oils*

Essential oils contain aromatic, partially water-soluble, inflammable, and volatile secondary metabolites that are soluble in organic solvents. Steam distillation is used to extract essential compounds from plant material. Essential oils include monoterpenes, sesquiterpenes, phenols, oxides, esters, aldehydes, and ketones. There are three main biosynthetic pathways that produce the most essential oils: mono- and diterpenes are created in the methyl-erithrytol pathway; sesquiterpenes are produced via the mevalonate pathway; phenylpropenes are formed via the shikimic acid pathway [66–68]. Essential oils were isolated from the aerial parts (leaves, stems, flowers) of *A. gypsophiloides* [49]. The chemical composition of the essential oil was qualitatively and quantitatively analyzed by GLC-MS and the yield of oil was 0.008% ($v/w$). Fifty-nine compounds were identified in total, accounting for 89.8% of the oil. Pulegone (**14**), *trans-p*-menthan-3-one (**15**), *trans*-verbenol (**16**), phytol (**17**), 2-isopropyl-5-methylcyclohexanone (**18**), and isopiperitenon (**19**) were the main constituents of *A. gypsophiloides* essential oil. The results presented that oxygen-containing monoterpenes made up the main fraction in the essential oil of *A. gypsophiloides*.

*6.5. Fatty Acids*

Fatty acids are lipid biomolecules that are present in all organisms and perform a variety of functions (component of biomembranes and lipid stores). They are used not only in the production of numerous food products, but also in soaps, detergents, and cosmetics [69]. No reports are available on the presence of fatty acids in *Allochrusa* species. For the first time, from the aerial parts of *A. gypsophiloides* using GLC-MS, several fatty acids were identified [49]. According to GLC-MS analysis, hexadecanoic acid (3.6%), tetradecanoic acid (1.1%), octadecanoic acid (0.5%), and linoleic acid (0.2%) were the main fatty acids in *A. gypsophiloides*.

*6.6. Polysaccharides, Pectins, and Hemicelluloses*

*Allochrusa* species are plants rich in polysaccharides. From the aerial organs (stems and leaves) and roots of *A. gypsophiloides* water-soluble polysaccharides, pectin substances, and hemicellulose were isolated. Arifkhodzhaev et al. [46] reported that the neutral polysaccharide of the roots of *A. gypsophiloides* was polydisperse (molecular weights from 1000 to 2000); the polysaccharide included galactose and glucose residues (5:1) with traces of mannose, arabinose, and rhamnose residues. Later, the same researchers [43] investigated the polysaccharides of the stems and leaves, which did not contain polysaccharides of the glucogalactan type that are characteristic for the roots of the plant.

The water-soluble polysaccharides of *A. gypsophiloides* are described as a mixture of low-molecular-weight glucogalactans. The results of the structural analysis of a glucogalactan from the underground part of *A. gypsophiloides* showed that the glucogalactan is a branched polysaccharide whose main chain is composed of $\alpha$-1→6-bound galactopyranose residues. C-2 atoms of the galactopyranose residues function as the branching points, and the reducing ends of the glucogalactan are both galactopyranose and glucopyranose residues [44]. The glucogalactan obtained from the *A. gypsophiloides* was partially acid hydrolyzed and produced four oligosaccharides. Two of them, branched tetra- and pentasaccharides, were obtained for the first time [45].

The polysaccharides obtained from the roots of *A. paniculata* have been studied by Arifkhodzhaev [16] and pectin substances, water-soluble polysaccharides, and hemicelluloses have been yielded and described. The principal neutral polysaccharide of *A. paniculata* was a glucoarabinogalactan. The quantity, structure, and monosaccharide composition of the neutral polysaccharide of *A. paniculata* were significantly different from those of the neutral polysaccharide of *A. gypsophiloides.* The water-soluble polysaccharides consisted of glucose, galactose, mannose, xylose, arabinose, and rhamnose residues in the following ratios: 5.0:11.0:1.0:3.5:4.4:4.0, respectively. The pectin substances within the neutral polysaccharide of *A. paniculata* contained galactose, glucose, mannose, arabinose, and rhamnose residues

(1.1:1.0:1.0:8.5:2.8 ratio, respectively). The results also showed that the hemicelluloses of the neutral polysaccharide were composed of galactose, glucose, mannose, xylose, arabinose, and rhamnose residues (1.0:2.1:1.6:3.6:3.2:tr.). In addition to the neutral saccharides mentioned above, all the polysaccharides of *A. paniculata* also had uronic acid units bonded within their structures [16].

### 6.7. Other Compounds

The species of *Allochrusa* have tested positive for alkaloids, lignins, and carbohydrates [48]. Lignins are polymeric compounds with complex structures. The natural lignin of *A. paniculata* has been studied by catalytic hydrogenolysis and the results showed that there are three forms of lignin structural units: syringyl, guaiacyl, and *p*-coumaryl, which is typical for plants [50].

## 7. Biological Activities

*Allochrusa* species are used for a variety of purposes in conventional medicine around the world; therefore, they have been studied for their different biological activities. Mostly, *Allochrusa* species have been used in Central Asian ethnomedicine for treating bronchitis, gastrointestinal, skin and genital conditions, as well as spleen, liver, and kidney disorders [24]. The biological activities, including adjuvant, cytotoxic, anti-inflammatory, hemolytic, antifungal, analgesic, antioxidant, and other effects, have been revealed in some studies. Most investigations of triterpene glycosides have been conducted without studying the nature of the active compounds. The findings of these investigations, including both positive and negative results, are given below.

### 7.1. Anti-Inflammatory Activity

Any damage of cells or tissues results in inflammation, often accompanied by pain [27,28,70,71]. Chronic diseases can develop as a result of the persistent exposure to these different stresses, which can include viruses, toxins, UV radiation, and pollution, among others. The components isolated from the *Allochrusa* species have never been studied for their anti-inflammatory effects. However, there is evidence that saponins have anti-inflammatory properties. In many studies, the mice paw model with histamine-induced acute inflammation was employed to determine edema development [57]. There were two methods to administer saponins: orally and intraperitoneally. In the first experiment, compound **5** (20 mg/kg and 50 mg/kg body weight), compound **6** (20 mg/kg and 50 mg/kg BW), indomethacin (20 mg/kg BW), and water were given orally to six groups of eight mice each (control). An hour after receiving the medications, each mouse received a subcutaneous injection of 0.05 mL 0.1% histamine in the right hind paw. Edema production was evaluated after the injection of histamine. The anti-inflammatory effect of compounds **5** and **6** was measured by the decrease in the index of edema compared to the control group, which is calculated as the percentage difference between the mass of the healthy and the inflamed paw, relative to the mass of the healthy paw. In contrast to the experiment with oral administration, saponins 5 and 6 had an anti-inflammatory effect that was generally dose dependent. Compound **6** failed to demonstrate any consistent anti-inflammatory activity in the trial based on oral dosing. The results of the experiments showed that compound **5** had stronger anti-inflammatory effects than compound 6 in the trial based on intraperitoneal dosing [58]. The ecdysteroid 11 isolated from *A. gypsophiloides* was studied for its anti-inflammatory activity by Tuleuov et al. [47]. In this screening, the anti-inflammatory effect of compound **11** was performed on male rats. Peritonitis (acute exudative response) was induced by injecting 1% acetic acid intraperitoneally (1 mL per 100 g of the rat body weight). Compound **2** decreased the exudation volume in rats by 24% at a dosage of 50 mg/kg body weight when compared to the control group. Saponins may exert corticomimetic properties or can inhibit phospholipase A2α (PLA2α); these properties could explain the anti-inflammatory activities [72].

### 7.2. Immunomodulatory Activity

Saponins from *A. gypsophiloides* (Algiox) can form immunostimulatory complexes (ISCOMs) that stimulate cellular immunity in mice as documented by Turmagambetova et al. [32]. As control preparations, ISCOMs were used containing the saponin Quil-A and micelles of H7N1 influenza virus glycoprotein antigens. The results demonstrated that a single-dose subcutaneous immunization with ISCOMs containing Algiox saponins was able to stimulate different phases of immunity (similar to ISCOMs containing the saponin Quil-A). A water-extractable fraction of saponins, Quil A, is derived from the plant *Quillaja saponaria*. When compared to immunization with micelles of H7N1 influenza virus glycoprotein antigens, the studied ISCOMs significantly increased the levels of IFN-, IL-2, IL-4, and IL-10 cytokines. The findings demonstrated that Algiox saponins could form immunostimulatory nanocomplexes with a similar immunostimulatory efficacy and structure to ISCOMs containing the saponin Quil-A. It has been proposed by the authors [32] that Algiox saponins obtained from *A. gypsophiloides* can be used to stimulate cellular and humoral immunity.

### 7.3. Adjuvant Activity

Saponins are known as effective adjuvant molecules. Extracts from *A. gypsophiloides* exhibited nonspecific activity compared to various antigenic preparations such as bovine serum albumin, the parainfluenza virus, and PG-3 [73]. Acanthophylloside (allochroside or a mixture of acanthophyllosides A-D) mediated the most potent adjuvant efficiency, which was similar or even superior to the other saponin preparations obtained from the "Merck", "Spofa", and "DVN" companies. The agent (allochroside) was also an effective adjuvant when used as an ingredient in the inactivated adsorbed vaccine for foot and mouth disease. It substantially enhanced its immunogenicity in cattle experiments. The data obtained in the course of the experiments showed that the administration of acanthophylloside to guinea pigs significantly activated the process of their antibody formation against immune bovine serum albumin. Based on the data they obtained, the authors concluded that saponins, the aglycones of which are represented by gypsogenin and quillaic acid, have a pronounced adjuvant activity [73]. Khatuntseva et al. [58] recently evaluated the adjuvant potential of triterpene glycosides **5** and **6** isolated from *A. gypsophiloides*. Test immunizations were carried out by using the keyhole limpet hemocyanin (KLH) protein carrier and synthetic vaccine neoglycoconjugate 3′SL-KLH (α-NeuAc-(2→3)-β-Galp-(1→4)-β-Glcp-KLH) on the basis of 3′SL (3′-sialyllactoside) ligands. The specific anti-3′SL IgM and IgG responses were measured after immunizing four groups of mice with the saponin of *Quillaja* bark, 40 μg of 3′SL-KLH along with 50 μg of compounds **5** and **6**, or without an adjuvant. Comparing the test vaccine formulation with the antigen alone in terms of the control vaccine formulation, a substantial specific response was seen for saponins **5** and **6**. The results demonstrated that high blood titers of IgM and IgG antibodies were recorded in the vaccination using compound **6** as an adjuvant, while the IgG level was less with the saponin from *Quillaja* bark. The levels of such antibodies in the saponin **5** experiment were rather modest. The authors concluded that antigen compound **6** demonstrated strong adjuvant qualities when combined with 3′SL-KLH and can thus be viewed as a potential component of vaccination formulations. The investigations of Turmagambetova et al. (2017) demonstrated that Algiox was capable of forming immunostimulatory nanocomplexes that were similar to ISCOMs containing the saponin Quil-A in terms of structure and immunostimulatory effectiveness. The adjuvant effect of the Algiox was studied in mice by subcutaneous injection with lipids and glycoproteins of the H7N1 influenza virus and ISCOMs containing saponins. The findings showed that H7N1 viral glycoprotein antigens, ISCOMs containing lipids, and saponins induced a high level of IgG and IgG2a serum antibodies. The level of the immune response and protective efficacy was greater than that induced by micelles containing glycoproteins combined with an aluminum hydroxide adjuvant, viral glycoproteins, or whole virions; the response was also similar to that elicited by ISCOMs containing Quil-A.

As a result of the investigations, Algiox was recommended as an additional source of effective adjuvants for both human and veterinary vaccines [32].

### 7.4. Hemolytic Activity

Triterpene saponins are amphiphilic substances that can interfere with biological membranes. Saponins with lipophilic moiety can complex cholesterol and related lipids inside biomembranes; the hydrophilic side chains bind to proteins on the cell surface. This tension leads to a cell lysis. It can be best demonstrated with red blood cells, which disrupt when exposed to saponins [27,28,70]. The hemolytic activity of a bacterium or molecule is its ability to destroy red blood cells, leading to the release of hemoglobin.

The researchers Bazhenova and Aliev [74] investigated the hemolytic index of the saponin (a mixture of seven paniculatosides) isolated from *A. paniculata*. The hemolytic activity of paniculoside was determined and paniculatoside has a pronounced activity equal to 1 in 5000. For the purified saponin Algiox, the investigation of in vitro hemolysis was performed by Turmagambetova et al. [32]. The Quil-A concentration that caused 50% hemolysis ($HD_{50}$) was 225 µg/mL, whereas the $HD_{50}$ for Algiox was 700 µg/mL. The Algiox preparation that was tested contained derivatives of quillaic acid [75]. An in vitro hemolysis investigation was conducted by Khatuntseva et al. [58] using compounds **5**, **6,** and *Quillaja* bark saponin (as a control). According to the results, the saponin from *Quillaja* bark induced 100% of the hemolysis at the dose of 5.5 µg/mL. The hemolytic activity of saponins **5** and **6** was substantially lower; 50% hemolysis was seen at doses of 11 to 18 µg/mL, respectively. Compounds **5** and **6** caused hemolysis of 85–95% at a concentration of 62.5 µg/mL. Reference saponins QS-17, 18, and 21 obtained from *Quillaja* bark induced hemolysis at doses of 7–25 µg/mL. The findings are in good agreement with expectations that bidesmosidic saponins (as in the case of compounds **5** and **6**) and the existence of glucuronic acid moiety at C-3 position of the aglycone have limited hemolytic activity [58].

### 7.5. Cytotoxic Activity

Saponins exhibit cytotoxic activities that are promoted by their membrane-disrupting activities described for hemolysis. The in vitro and in vivo toxicity of the Algiox saponin from *A. gypsophiloides* was experimentally evaluated [32]. In comparison to the saponin Quil-A, which was employed as a reference, the saponins isolated from *A. gypsophiloides* displayed lower cytotoxicity in tests performed with model cell cultures. The cytotoxic concentrations ($IC_{50}$) of Quil-A were 21 µg/mL and 28 µg/mL in macrophages and MDCK (Madin–Darby canine kidney) cells, respectively. The $IC_{50}$ values of Algiox in macrophages and MDCK cells were 103.5 µg/mL and 93.75 µg/mL, respectively. The cytotoxicity of the essential oil isolated from the aerial parts (leaves, stems, flowers) of *A. gypsophiloides* was determined in human colon adenocarcinoma (HT-29) and prostate cancer (PC-3) cells by MTT assay [49]. The results showed a moderate cytotoxicity against PC-3 ($IC_{50}$ = 89.9 ± 2.01 µg/mL) and HT-29 cells ($IC_{50}$ = 43.6 ± 2.38 µg/mL). In a different study, biologically active compounds were extracted from the aerial parts of *A. gypsophiloides* using methanol, chloroform, and water as solvents [76]. The water extract had mild cytotoxic effects on leukaemia cell lines (CEM/ADR-5000: IC50 = 31.9 g/mL), leukemic lymphoblasts (CCRF-CEM), and breast cancer cells (MCF-7: IC50 = 145.9 g/mL).

### 7.6. Analgesic Activity

Tuleuov et al. [47] studied the analgesic activities of 3-epi-2-deoxyecdysone (**11**) isolated from *A. gypsophiloides*. The peritoneal chemical irritation test (also known as the acetic acid writhing test) in white outbred mice was used to examine the analgesic effects of the compound **11**. The mice were injected with 0.1 mL/10 g BW intraperitoneally with an acetic acid (0.75%) solution. The samples were administered intragastrically at a dose of 50 mg/kg BW before the acetic acid injection. The "writhes" were measured immediately following administration and for a period of 30 min. Tests were conducted using a concentration of

50 mg/kg BW of the reference painkiller diclofenac. The results of the test showed that compound **11** exerted an analgesic effect.

### 7.7. Antifungal Activity

Triterpene glycosides also interfere with biomembranes of fungi and thus have often shown antifungal properties [27,28,70]. Compounds **5** and **6** were tested against four fungal cultures: basidiomycetous (*Cryptococcus terreus*, *Filobasidiella neoformans*) and ascomycetous (*Saccharomyces cerevisiae*, *Candida albicans*) yeasts [58]. The results demonstrated that compounds **5** and **6** exhibited an antifungal effect against all the tested yeasts, especially the therapeutically significant *C. albicans* and *F. neoformans*. Growth-inhibiting tests revealed that saponins **5** and **6** had stronger antifungal effects at a pH of 4.0 than at a higher pH. At pH 7.0, the tested substances had no effect against *C. albicans* and *F. neoformans*. However, against these two strains both saponins were antifungal at pH 4.0. Notably, *S. cerevisiae* was resistant to compound **5** at either pH level.

### 7.8. Antioxidant Activity

The DPPH (2,2-diphenyl-1-picrylhydrazyl) and ABTS (2,2′-azino-bis (3-ethylbenzothia zoline-6-sulfonic acid) diammonium salt) assays are a simple, rapid, and sensitive method for evaluating free radical scavenging ability. Our results demonstrated that the DPPH free radical scavenging ability of the essential oil of *A. gypsophiloides* was moderate ($IC_{50}$ = 456.4 $\pm$ 5.39 μg/mL) [49]. According to the $IC_{50}$ value (745.0 $\pm$ 6.87 μg/mL), a weak ABTS scavenging activity was observed for the essential oil.

### 7.9. Acute Toxicity

The toxicity of the mixture of paniculatosides (paniculatoside) was investigated on white mice that had been treated with saponins (intravenously, subcutaneously, and orally) [74]. With an intravenous administration of paniculatoside in doses of 5–20 mg/kg BW, mice showed an increase in motor activity, followed by depression, and the mice huddled together. At high doses, mice died on the first day, and at low doses some mice died on the second and third days after treatment. Therefore, observations were carried out for 8 days. The following $LD_{50}$ values were obtained: the $LD_{50}$ value for the intravenous administration was 11.1 mg/kg, in the case of subcutaneous administration it was 77.5 mg/kg, and for oral administration 2550 mg/kg. Thus, paniculoside is a relatively toxic drug. Its toxic effect is reduced by subcutaneous and oral administration, which is typical for saponins [74]. The acute toxicity study of two triterpene glycosides (**5** and **6**) isolated from the roots of *A. gysophiloides* was carried out in vivo with male mice [58]. After a single oral or intraperitoneal dose, the corresponding $LD_{50}$ values were calculated: The $LD_{50}$ for oral administration ranged between 252 $\pm$ 57 mg/kg for compound **5** and 304 $\pm$ 55 mg/kg for compound **6**. The $LD_{50}$ for the intraperitoneal administration of compounds **5** and **6** was 5.4 $\pm$ 2.8 mg/kg BW and 15.1 $\pm$ 5.6 mg/kg BW, respectively. The strong differences in $LD_{50}$ values between oral and intraperitoneal experiments indicated that the saponins were not being absorbed in the intestines. In in vivo tests, the toxicity of Algiox was evaluated [32]. Within the study, the toxicity was evaluated based on lethality, weight, and hair loss. For chickens and chicken embryos, the Quil-A $LD_{50}$ was 225 μg/animal, while for mice, it was 187.5 μg/animal. Algiox's $LD_{50}$ in mice was found to be 650.4 μg/animal, whereas it was 850 μg/animal in chickens and chicken embryos. Lethality experiments performed on mice, chicken embryos, and chickens revealed that the assessed Algiox saponins were not toxic in the range of concentrations used. The standard threshold dose for vaccines is 15.0 μg per animal, which is at least 40 times less than the $LD_{50}$ calculated for the Algiox saponins.

### 7.10. Other Activities

Bazhenova and Aliev [74] investigated several pharmacological parameters (the effect on the central nervous system, blood pressure, heart arrhythmia, smooth muscle, diuresis) of the saponins isolated from *A. paniculate* (a mixture of seven paniculatosides) in rats.

In doses of 1–10 mg/kg body weight, paniculoside briefly decreases blood pressure by 40–50%, followed by its restoration. There are no significant changes in respiration. In experiments with the in situ rat heart, paniculatoside at a concentration of 3 mg/kg body weight increased the amplitude (by 14%) and slightly slowed down the heart rate. With ECG research, the R-R interval increased by 31%. In doses of 3–5–10 mg/kg BW, the drug did not relieve or prevent the development of calcium chloride arrhythmia caused by the intravenous administration of calcium chloride (270 mg/kg). The drug at a concentration of 5 mg/kg body weight significantly, and at a concentration of 20 mg/kg almost completely, suppressed the components of the orientation reaction in white mice. At a concentration of 50 mg/kg, paniculatoside does not affect the sleep of white mice induced by chloral hydrate (300 mg/kg), hexanal (75 mg/kg), and nembutal (60 mg/kg). At the same time, at the concentration 50 mg/kg, it remarkably lengthened the latent period of corazole seizure onset. Paniculotoside did not have a ganglioblocking effect and did not change the blood pressure response caused by the intravenous administration of acetylcholine (0.5 μg/kg BW) and adrenaline (10 μg/kg BW). Paniculotoside enhanced the action of histamine (0.5–1 μg/kg BW) both in the acute experiment and in an isolated segment of the intestine of rats, which occurs 30–60 min after the administration of the drug and lasts 1.5–2 h. In experiments on rats, the drug in doses of 10–20 mg does not have a diuretic effect. It briefly lowers blood pressure and enhances the effect of histamine when administered exogenously [74]. Saponin isolated from the roots of *A. gypsophiloides* decreased the convulsive and toxic effects of corazole, potentiated the convulsive effect of strychnine, antagonized the narcotic effect of chloral hydrate, and increased diuresis in mice [77]. We studied the anthelmintic effect of the methanol, chloroform, and water extracts of *A. gypsophiloides* on *Caenorhabditis elegans*. Antihelmintic tests revealed that samples at the tested concentration of 500 μg/mL did not kill *C. elegans* [78].

## 8. Conclusions and Future Perspectives

This review revealed that *Allochrusa* species possess diverse chemical compounds and biological activities and people from Central Asia use these plants to treat a wide spectrum of human diseases. *Allochrusa* species as food and medicinal plants have always been used by people and the impact of plant-based foods and their bioactive secondary metabolites on human health and wellbeing is undoubted. It is necessary to collect this ancient knowledge through official documentation and keep it for future research. The genus *Allochrusa* is native to the mountains and deserts of Central and South–Central Asia. Apparently, *Allochrusa* is a good source of bioactive molecules. Therefore, in the next stage of research, the components of *Allochrusa* species should be subjected to wider phytochemical and pharmacological analyses. The phytochemistry of *Allochrusa* has not been exhaustively studied, not even in the well-studied species of *Allochrusa*. Phytochemical studies on *A. gypsophiloides* are scarce and were mostly conducted in the 1960s–1980s of the last century. Certainly, it would be worth investigating all species for new bioactive molecules. It was established that triterpene glycosides serve as chemotaxonomic markers for the characterization of *Allochrusa* species. In recent years, natural saponins of this genus have been found to have significant adjuvant, cytotoxic, anti-inflammatory, hemolytic, antifungal, analgesic, antioxidant, and other effects. Therefore, in the next stage of research, the components of *Allochrusa* species should be subjected to wider phytochemical and pharmacological analyses. Additional chemical analyses of other metabolites using modern analytical methods and pharmacological studies will complete the knowledge of this genus. In addition to their possible use as detergents, saponins have a wide range of other potential applications, including those in the food, cosmetics, and pharmaceutical industries. The saponins from *A. gypsophiloides* were recommended as an additional source of highly effective adjuvants for vaccines. Some *Allochrusa* species are rare and endangered in nature. The thickets of *Allochrusa* have been significantly depleted in recent years, which was the reason for including selected species in the Red Data Books. To meet the needs of the pharmaceutical and food industries for raw materials, it is necessary to cultivate these

plants. In this regard, it is necessary to determine the optimal growing conditions of the economically relevant species.

**Author Contributions:** R.M. conceptualization, draft preparation, writing; V.K. supervising, review and editing; M.Š. investigation, methodology, revising; N.Z.M. writing, resources; M.W. writing, review and editing. All authors have read and agreed to the published version of the manuscript.

**Funding:** This research received no external funding.

**Institutional Review Board Statement:** Not applicable.

**Informed Consent Statement:** Not applicable.

**Data Availability Statement:** All the data are available in the manuscript.

**Acknowledgments:** The authors would like to thank the Ministry of Higher Education, Science and Innovation of the Republic of Uzbekistan for financial support. This research was supported by the Scientific Grant Agency of the Slovak Republic VEGA (Project number 2/0118/23). This publication was created with the support of the Operational Program Integrated Infrastructure for the project: Development of products by modification of natural substances and study of their multimodal effects on COVID-19, ITMS: 313011ATT2, co-financed by the European Regional Development Fund.

**Conflicts of Interest:** The authors declare no conflict of interest.

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
