# Peer review of "The Genus Allochrusa: A Comprehensive Review of Botany, Traditional Uses, Phytochemistry, and Biological Activities"

_diversity, doi:10.3390/d15040574_

Round 1

Reviewer 1 Report

Dear Author, I reviewed the manuscript (diversity-2312264) entitled A Comprehensive Review of the Genus Allochrusa (Caryophyllaceae): Distribution, Traditional Uses, Phytochemical Composition, and Biological Activities. This manuscript presents relevant information about Allochrusa bioactivity. However, some sections of the submitted data can be improved. For this reason, I consider that this manuscript needs minor changes to be considered for publication in this journal. 

Additional comments.

Highlight the advantages of using  Allochrusa plants as medicinal plants.

Check paragraphs extension in this manuscript.

Include figure description in molecules structures included in this manuscript.

Compare the obtained findings with similar reviews where similar plant genera were analyzed. 

Include future trends to keep working with the obtained data. 

Try to conclude with a general statement of the most relevant part of this study.

Author Response

Dear Respected Reviewer,

We appreciate your valuable comments that aim to improve the quality of our work. Thank you for your time and job. Changes were made and tracked throughout the whole manuscript (details are addressed below).

(Reviewer 1)

Comments and Suggestions for Authors

Dear Author, I reviewed the manuscript (diversity-2312264) entitled A Comprehensive Review of the Genus Allochrusa (Caryophyllaceae): Distribution, Traditional Uses, Phytochemical Composition, and Biological Activities. This manuscript presents relevant information about Allochrusa bioactivity. However, some sections of the submitted data can be improved. For this reason, I consider that this manuscript needs minor changes to be considered for publication in this journal. 

# Thank you for appraising our work, we will pay all attention to revise all the required minor changes

Additional comments.

Highlight the advantages of using Allochrusa plants as medicinal plants.

#Thanks for your suggestions, correction is done and the advantages of using Allochrusa plants as medicinal plants.

Check paragraphs extension in this manuscript.

#Thanks for your suggestions, the paragraphs length were adjusted.

Include figure description in molecules structures included in this manuscript.

# The structures of the major volatile compounds were included in the Table 2 as other reviewer asked.

Compare the obtained findings with similar reviews where similar plant genera were analyzed. 

# Thank you very much for the constructive comment, we have tried our best to allocate the closest genus taxonomically to Allochrusa which was Acanthophyllum. Unfortunately, we could not traced any reviews for this genus.

Include future trends to keep working with the obtained data. 

# The required information was included in “Conclusion and future perspective” part page 9 line 542- 565.

Try to conclude with a general statement of the most relevant part of this study.

# The required data were added, and manuscript corrected.

Best regards,

Authors

Reviewer 2 Report

Nice and solid work on the phytochemistry of one genus (Allochrusa). The review is well done and referees to the most recent literature in chemistry. It also introduces in the scientific circulation of some older literature in Russian, which is important from the perspective of ethnobiology. My suggestion would be to provide the original names of the sources in Russian along with he translation of the content. Now for example reference 20 has a mixture of translation and transliteration and the such source is difficult to trace. 

Author Response

Dear Respected Reviewer,

We appreciate your valuable comments that aim to improve the quality of our work. Thank you for your time and job. Changes were made and tracked throughout the whole manuscript (details are addressed below).

(Reviewer 2)

Comments and Suggestions for Authors

Nice and solid work on the phytochemistry of one genus (Allochrusa). The review is well done and referees to the most recent literature in chemistry. It also introduces in the scientific circulation of some older literature in Russian, which is important from the perspective of ethnobiology. My suggestion would be to provide the original names of the sources in Russian along with the translation of the content. Now for example reference 20 has a mixture of translation and transliteration and the such source is difficult to trace. 

# We express our sincere gratitude for the careful analysis of our manuscript. We are sorry for that unintended mistake, and the required corrections are done.

Best regards,

Authors

Reviewer 3 Report

A Comprehensive Review of the Genus Allochrusa (Caryophyllaceae): Distribution, Traditional Uses, Phytochemical Composition and Biological Activities

Reviewer comments

The title of the article needs to be corrected, it should be more concise and relevant.

The section “Materials and Methods” should be included in the Introduction section.

Section 5. “Traditional uses” would fit more as the last section after 7. “Biological activities”

The "Conclusions" are missing the content to match the paper's title and need improvement.

Some experimental descriptions are too detailed and do not significantly relate to the title of the paper, text in lines 340-350, 391-402, 430-436.

The list of references seems too short for a review paper, which is why the paper is missing a broader view and context.

Other comments and suggestions

Line 87-90,  sepals? or petals?

Table 1 , move the “Species” column to the first position on the left

Table 2, move the “Species” (“Plant resources”) column to the first position on the left

Title 6. “Phytochemical studies” change to “Phytochemical compounds”

The formulas of essential oils have to be included in table 2

Line 325, please change “folk medicine” to “ethnomedicine”

Author Response

Dear Respected Reviewer,

We appreciate your valuable comments that aim to improve the quality of our work. Thank you for your time and job. Changes were made and tracked throughout the whole manuscript (details are addressed below).

 (Reviewer 3)

Comments and Suggestions for Authors

Reviewer comments

The title of the article needs to be corrected, it should be more concise and relevant.

# Thanks for your suggestions, the title was changed as required.

The section “Materials and Methods” should be included in the Introduction section.

# Thanks for your suggestions, but we think method part of the review should be given separately from introduction. The Methods part of a review paper mainly characterize how the relevant literature was selected (which database, which search terms..) and how it was then analyzed and summarized. Many reviews published in Diversity (https://doi.org/10.3390/d15020138, https://doi.org/10.3390/d15030467, https://doi.org/10.3390/d15030409, etc.) gave Methods and materials part separately from Introduction. 

Section 5. “Traditional uses” would fit more as the last section after 7. “Biological activities”

# In section 5. "Traditional uses" contains information not only about Biological activities but also about their use in different branches of industry (e.g., cosmetics, textiles, food, etc.) and we gave the "Medicinal Uses" part separately.

The "Conclusions" are missing the content to match the paper's title and need improvement.

# Thanks for your suggestions, the Conclusion part improved.

Some experimental descriptions are too detailed and do not significantly relate to the title of the paper, text in lines 340-350, 391-402, 430-436.

# Thank you for your comment, as the title of the manuscript include biological activities of the species from the genus Allochrusa in general, we have collected all the possible and relevant sources related to the topic to present a comprehensive review that might guide the researchers worldwide. Since the topic biological is quite general some details are required to cover all the aspects.

The list of references seems too short for a review paper, which is why the paper is missing a broader view and context.

# As we have written in the methodology part, we have searched all the possible databases with different keywords combinations to retrieve all the possible references but we noticed that the number of researches in Allochrusa are not too many. There are only 8 plant species in this genus, and only 3-4 have been studied. More research is required on the phytochemical constituents and biological activities of this genus. And we think our review will help future researchers discover new therapeutic agents as the species of this genus possess various interesting pharmacological potentials.

Other comments and suggestions

Line 87-90,  sepals? or petals?

# revised.

Table 1 , move the “Species” column to the first position on the left

# done.

Table 2, move the “Species” (“Plant resources”) column to the first position on the left

# done.

Title 6. “Phytochemical studies” change to “Phytochemical compounds”

# changed.

The formulas of essential oils have to be included in table 2

# done.

Line 325, please change “folk medicine” to “ethnomedicine”

# changed.

Best regards,

Authors